

# Soil-atmosphere exchange of carbonyl sulfide in Mediterranean citrus orchard

Fulin Yang#, Rafat Qubaja, Fyodor Tatarinov, Rafael Stern, and Dan Yakir*
Earth and Planetary Sciences, Weizmann Institute of Science, Rehovot 76100, Israel
#*Present address*: College of Animal Sciences, Fujian Agriculture and Forestry
University, Fuzhou 350002, China
*Correspondence:* Dan Yakir; email: dan.yakir@weizmann.ac.il





**Abstract:**

Carbonyl sulfide (COS) is used as a as a tracer of $CO_2$ exchange at the ecosystem
and larger scales. The robustness of this approach depends on knowledge of the soil
contribution to the ecosystem fluxes, which is uncertain at present. We assessed the
spatial and temporal variations of soil COS and $CO_2$ fluxes in the Mediterranean citrus
orchard combining surface flux chambers and soil concentration gradients. The spatial
heterogeneity in soil COS exchange indicated net uptake below and between trees of
up to -4.6 pmol m$^{-2}$ s$^{-1}$, and net emission in exposed soil between rows, of up to +2.6
pmol m$^{-2}$ s$^{-1}$, with weighted mean uptake values of -1.10 ± 0.10 pmol m$^{-2}$ s$^{-1}$. Soil
COS concentrations decreased with soil depth from atmospheric levels of ~450 to ~100
ppt at 20 cm depth, while $CO_2$ concentrations increased from ~400 to ~5000 ppm. COS
flux estimates from the soil concentration gradients were, on average, -1.02 ± 0.26 pmol
m$^{-2}$ s$^{-1}$, consistent with the chamber measurements. A soil COS flux algorithm driven
by soil moisture and temperature (5 cm depth) and distance from the nearest tree, could
explain 75% of variance in soil COS flux. Soil relative uptake, the normalized ratio of
COS to $CO_2$ fluxes was, on average -0.37 and showed a general exponential response
to soil temperature. The results indicated that soil COS fluxes at our study site were
dominated by uptake, with relatively small net fluxes compared to both soil respiration
and reported canopy COS fluxes. Such result should facilitate the application of COS
as a powerful tracer of ecosystem $CO_2$ exchange.

**Keywords:**


Carbonyl sulfide; COS; OCS; soil gas exchange; ecosystem gas exchange; tracer of
carbon fluxes.





## 1. Introduction


Carbonyl sulfide (COS) is a Sulphur-containing analogue of $CO_2$ that is taken up
by vegetation following a similar pathway to $CO_2$, ultimately hydrolyzed in an
irreversible reaction with carbonic anhydrase. It therefore holds great promise for
studies of photosynthetic $CO_2$ uptake (Asaf et al., 2013;Berry et al., 2013;Wehr et al.,
2017;Whelan et al., 2018). One of the difficulties in the application of COS as a tracer
for photosynthetic $CO_2$ uptake is the unknown non-leaf contributions the net ecosystem
COS flux. There are reports of significant soil fluxes, indicating both uptake and
emissions (Kesselmeier et al., 1999;Kuhn et al., 1999;Masaki et al., 2016;Seibt et al.,
2006;Yang et al., 2018;Yi et al., 2007). Although soil COS exchanges were often
considered small compared to plant uptake (e.g., Whelan et al., 2013;Yang et al., 2018),
this was not always the case. Significant soil COS emissions have been found in
wetlands and anoxic soils (Li et al., 2006;Whelan et al., 2013), and in senescing
agricultural fields and high temperatures (Liu et al., 2010;Maseyk et al., 2014), or under
drought conditions and in response to UV radiation (Kitz et al., 2017). Event for the
same soil, COS fluxes could show large variations and both uptake and emission with
sensitivities to soil moisture, and ambient COS concentrations (Bunk et al.,
2017;Kaisermann et al., 2018). These studies also assessed the response of COS
exchange to environmental controls, e.g. soil moisture and temperature and solar
radiation.
Soil COS exchange has often been measured by incubations in the lab (e.g., Bunk
et al., 2017;Kesselmeier et al., 1999;Liu et al., 2010;Van Diest and Kesselmeier, 2008),
and by static or dynamic chambers in the field (e.g., Berkelhammer et al., 2014;Kitz et
al., 2017;Sun et al., 2018;Yi et al., 2007;Mseyk et al., 2014), and using models (e.g.,
Ogée et al., 2016;Sun et al., 2015;Whelan et al., 2016). In spite of these efforts, more
field measurements of soil COS exchange are clearly needed as a basis for elucidating
underlying mechanism, as well as obtaining better quantitative record of the possible
range of soil COS fluxes under natural conditions. The objective of this study was to
apply dynamic chambers measurements, constrained by simultaneous soil gradient
method to assess the spatial and temporal variations soil COS and $CO_2$ fluxes in a citrus



orchard ecosystem where contrasting soil microsite conditions occur.

## 2.    Materials and methods

### 2.1  Field site

The experiment was conducted in an orchard in Rehovot, Israel (31°54′ N, 34°49′
E, 50 m, asl) in 2015 and 2016. The orchard is a plantation of lemon trees (*Citrus*
*limonia Osbeck*), with 5 m distance between rows and 4 m between trees. Mean annual
air temperature at the site is 19.7 °C, and mean annual precipitation is 537 mm. Most
of the precipitation (82%) falls in November to February with no rain during June to
October. A trickle irrigation system was used from May to September with the standard
irrigation plan of the orchard management. The soil in the area is red sandy soil with an
average bulk density of 1.6 kg m$^{-3}$ (Singer, 2007).

### 2.2  Quantum cascade laser measurements

We used the commercially available quantum cascade laser (QCL) system
(Aerodyne Research, Billerica, MA) with tunable laser absorption spectrometer (Model:
QC-TILDAS-CS) to measure COS, $CO_2$, and water vapor concentrations
simultaneously. The device was installed in a mobile lab, described in the Asaf et al.
(2013). COS is detected at 2050.40 cm$^{-1}$ and $CO_2$ at 2050.57 cm$^{-1}$ at a rate of 1 Hz.
The instrument was calibrated using working reference compressed air tank that was
used for inter-comparison with the NOAA GMD lab (Boulder CO).

### 2.3  Soil chamber flux measurements

Custom-made stainless-steel cylindrical chamber of 177 cm$^2$ directly inserted into
the soil (~5 cm) was used, as previously described (Berkelhammer et al., 2014;Yang et
al., 2018). The chamber air and ambient air flows were pumped to the QCL analyzer
through two 3/8-inch diameter Decabon tubing. Flow rate was maintained at 1.2 L min$^{-1}$
and repeatedly cycled with 1 min instrument background (using $N_2$ zero gas), 9 min
ambient air flow, and 10 min chamber air sample. Three different soil sites were used
with distance of 3.20, 2.00 and 0.25 m away from a tree trunk, that represented sampling



sites between rows (BR), between trees (BT) and under tree (UT). Each sampling site
was measured continuously for 24 hours and cycled between sites for the duration of
the campaign. Four measurement campaigns were carried out during 5th~9th August
2015; 25th~28th December 2015; 5th~9th May 2016; 28th~31th July 2016.
Gas exchange rates, $F_c$, were calculated according to:
$$F_c = \frac{Q}{A} \times \left( \Delta C_{sample} - \Delta C_{blank} \right) \qquad (1)$$

where $Q$ is the chamber flush rate in mol s$^{-1}$; $A$ is the enclosed soil surface in m$^2$; $\Delta C$ is
the gas concentrations difference between chamber air and ambient air in pmol mol$^{-1}$
for COS and µmol mol$^{-1}$ for $CO_2$ under sampling, and blank reference treatments (using
the same chamber placed above a sheet of aluminum foil before and after measurement
at each site. Hereafter, the soil fluxes are reported in pmol m$^{-2}$ s$^{-1}$ and µmol m$^{-2}$ s$^{-1}$ for
COS and $CO_2$, respectively. Soil relative uptake (SRU) is used to characterize the
relationship between soil $CO_2$ and COS fluxes, was estimated from the normalized ratio
of COS to $CO_2$ and uptake (negative values), or emission (positive values) fluxes
(Berkelhammer et al., 2014):
$$SRU = \frac{F_{COS_{soil}}}{[COS]} \Big/ \frac{F_{CO_{2soil}}}{[CO_2]} \qquad (2)$$


**2.4  Soil concentration profile measurements**
Four campaigns of soil concentration profile measurements were carried out
during 1st~2nd March; 20th~26th April; 10th May; 22nd~28th June of 2016. The trace
gas at five soil depths of 0, 2.5, 5.0, 10, 20 cm was sampled at each of the three
microsites, BR, BT and UT.
Four Decabon tubing were inserted into the soil at the different depth and connect
directly to the QCL. At least one day after insertion, soil air was sampled with flow rate
of 400 ml min$^{-1}$, in cycles of 1 min instrument background-3 min surface air (depth 0)-
5 min sampling of a depth point in the profile-1 min surface air. Each cycle was,
therefore, 10 min. Five complete sets of cycles including the four soil depths and
surface air were repeated for each site.





Assuming that in the selected measurement sites, soil trace gas is only
transported by diffusion, soil COS and $CO_2$ fluxes estimated based on the Fick's first
law according to:
$$F = -D_s \frac{dC}{dz} \qquad (3)$$

where $F$ is the upward or downward gas flux (pmol m$^{-2}$ s$^{-1}$ for COS and µmol m$^{-2}$ s$^{-1}$
for $CO_2$); $D_s$ is the effective gas diffusion coefficient of the relevant gas species in the
soil (m$^2$ s$^{-1}$); $C$ the trace gas concentration (mixing ratio, converted from the measured
mole fractions); $z$ is the soil depth (m).
The Penman (1940) function was used to describe the soil diffusion coefficient
($D_s$) as in Kapiluto et al. (2007):
$$D_s = D_a \left( \theta_s - \theta \right) \sqrt{\frac{T_s + 273.15}{298.15}} \qquad (4)$$

where $\theta_s$ is the soil saturation water content and $\theta$ is the measured soil volumetric water
content. $D_a$ is the trace gas diffusion coefficient in free air, which varied with
temperature and pressure, given by
$$D_a = D_{a0} \left( \frac{T + 273.15}{293.15} \right)^{1.75} \left( \frac{P}{101.3} \right) \qquad (5)$$

where $D_{a0}$ is a reference value of trace gas diffusion coefficient at 293.15 K and 101.3
kPa, given as $1.24 \times 10^{-5}$ m$^{-2}$ s$^{-1}$ for COS (Seibt et al., 2010) and $1.47 \times 10^{-5}$ m$^{-2}$ s$^{-1}$
for $CO_2$ (Jones, 1992); $T$ is temperature (ºC), and $P$ is air pressure (kPa).

**3.  Results**
**3.1 Variations in soil COS flux**
Soil COS fluxes showed significant heterogeneity at both the spatial (microsites)
and temporal (seasonal), as well as interactions between microsite and season (Figure
1, Table 1). Overall, the hourly soil COS flux varied from -4.6 to +2.6 pmol m$^{-2}$ s$^{-1}$,
with mean value of -1.10 ± 0.10 pmol m$^{-2}$ s$^{-1}$. On the spatial scale, the COS fluxes
showed systematically uptake under trees (UT), moderate uptake and some emissions
between trees (BT) and relatively more emission in the exposed area between rows





(BR), with diurnal mean values across seasons of -3.00 ± 0.10, -0.43 ± 0.13 and 0.13 ±
0.11 pmol m$^{-2}$ s$^{-1}$, respectively.

On the diurnal time-scale, soil COS flux were generally higher in the afternoon

(peaking around 15:00~16:00 hours), declining at night and early morning (Fig. 1). On
the seasonal time scale, soil COS fluxes showed both changes in rates and shifts from
net uptake to net emission, with interactions between site and season (Table 1). In the
UT sites where only COS uptake was observed, the highest rates were observed in
winter and peak summer (December and Auguest) with diurnal mean rates of nearly 4
pmol m$^{-2}$ s$^{-1}$, and more moderate uptake rates, around 2 pmol m$^{-2}$ s$^{-1}$, in spring and early
summer (May and July; Fig. 1). In the BT sites, significant COS uptake of ~-2.5 pmol
m$^{-2}$ s$^{-1}$ was observed in winter, but net fluxes were near zero in other times, with some
afternoon emission in summer. In the exposed BR sites, minor uptake (less than 1 pmol
m$^{-2}$ s$^{-1}$) was observed in spring and early summer, but consistent emission in peak
summer, with diurnal mean values of nearly 2 pmol m$^{-2}$ s$^{-1}$.

**3.2 Effects of moisture and temperature**

During the hot summer (August 2015 and July 2016), differences in microsite soil

water content ($\theta$) were most distinct, with $\theta$ of nearly 30% in the UT sites (associated
with drip irrigation), but ~19% and ~12% in the BT and BR sites. Correspondingly, the
UT sites had significant COS uptake of about -3 pmol m$^{-2}$ s$^{-1}$ while the other sites
showed emission of about +1 pmol m$^{-2}$ s$^{-1}$ (Table 1). In winter (December), $\theta$ in the
three sites was similar ~25% and all sites showed soil COS uptake, but with clear
gradient of -3.9, -2.5 and -0.7 pmol m$^{-2}$ s$^{-1}$ in the UT, BT and BR sites, respectively
(Table 1). On average, soil COS fluxes showed non-linear increase in uptake with
increasing $\theta$, but it seems that this response may saturate at about $\theta$ of 25% and uptake
rates of ~3.9 pmol m$^{-2}$ s$^{-1}$ (Fig. 2). The best fit line to the data presented in Fig. 2 also
indicate that in dry soil with $\theta$<15% soil COS emission can be expected.

The response of soil COS fluxes to soil temperature varied among the three

measurement sites (Fig. 3). The BT and BR sites showed a near linear response with a
shift from uptake to emisson around 25 °C. In the shaded and moist UT site, COS uptake





was always significant ranging between -4 to -1 pmol m$^{-1}$ s$^{-1}$ with relatively low
temperature sensitivity, and with lowest mean uptake rates around 20 °C.

Pearson product-moment correlation analysis results showed that hourly soil COS

flux was significantly related to soil mositure and temperature (at the 0.001 level), and
the soil mositure had a stronger environmental controls on the soil COS flux ($r$=-0.77),
compared with soil temperature ($r$=+0.45).

Comprehensive assessment of the effects of soil mositure ($\theta$), temperature ($Ts$) and

distance away from tree trunk ($d$) , hourly soil COS flux ($F_{COS}$) could be fitted to a three
parameters exponential model, which could explain 75% of the variation in soil COS
flux (Eq. 6).

$$F_{COS} = 8.91\exp(0.01Ts - 0.01\theta + 0.09d - 0.33) - 8.86, \ R^2 = 0.75 \qquad (6)$$


**3.3 COS flux estimates from soil concentration gradients**

The average soil concentration gradient of COS and $CO_2$ for the four campaigns

is shown in Fig. 4. COS concentrations decreased with soil depth, with the opposite
trend for $CO_2$, consistent with the results reported above of soil surface COS uptake
and $CO_2$ emission at our orchard site. COS concentrations at depth of 2.5 cm was on
average 314 ppt, and about one-third lower than the mean surface, ambient, value of
460 ppt. The lowest COS concentration at depth of 20 cm (166 ppt) was almost one-
third of that at the soil surface. An exponential and a linear equations provided
reasonable fit to the changes in soil COS and $CO_2$ concentrations, respectively, as a
function of depth ($z_{soil}$):
$$[COS] = 283.5\exp(-0.2z_{soil}) + 169.9, \ R^2 = 0.99$$
$$[CO_2] = 122.2z_{soil} + 558.5, \ R^2 = 0.99$$
$\qquad (7)$

In terms of individual site and campaign, all profiles except for BR in summer

(June) showed the general trend of decreasing [COS] and increasing [$CO_2$] with depth,
with the steepest gradient at the top 5 cm (Fig. 4). In the BR microsite in summer, $CO_2$
profile was shallow, consistent with the low respiration (see July BR in Table 1). But a
decrease in COS concentration toward the surface, with surface value lower than the



next two soil depth points (Fig. 5J), was consistent with COS emission at that time (July
BR in Table 1).
As noted above, the profile data generally exhibited the steepest gradient at the top
few cm of the soil, indicating that the dominating COS sink (and likely also the $CO_2$
source) was located at shallow depth. We therefore used the gas concentration
difference at two shallowest depths ($z_1 = 0$ and $z_2 = 2.5$ cm) to provide an approximation
of the fluxes to and from the soil, to constrain the more extensive chamber
measurements. The COS diffusion coefficient, $D_s$, was estimated for each campaigns
(see Methods), indicating low $D_s$ value in the UT site in June and July ($D_s = 2.55$ mm$^2$
s$^{-1}$), associated with the drip irrigation and the high soil water content, and high values
in the dryer soils ($D_s = 5.57$ mm$^2$ s$^{-1}$), with an average COS diffusion coefficient of 4.40
$\pm 0.29$ mm$^2$ s$^{-1}$. The soil COS flux estimates using the gradient method is reported in
Table 2. COS flux varied between -2.10 to +1.55 pmol m$^{-2}$ s$^{-1}$ with a mean value of -
$1.0 \pm 0.26$ pmol m$^{-2}$ s$^{-1}$ during the measurement periods, consistent with the mean value
of -1.10 $\pm$ 0.10 pmol m$^{-2}$ s$^{-1}$ reported above for the chamber measurements. Also in
agreement with the chamber measurements, fluxes at UT and BT always showed COS
uptake, with generally higher values in spring (March) than in summer (May-June),
while the BR data indicated change from uptake in spring (March-April, with -1.3 to -
1.6 pmol m$^{-2}$ s$^{-1}$) to emission in June (+1.6 pmol m$^{-2}$ s$^{-1}$).

**3.4 Soil relative uptake**
Soil was always a source of $CO_2$ due respiration (combined autotrophic and
heterotrophic respiration). Soil $CO_2$ flux rates varied both spatially and temporally in
similar patterns to those of COS, and with overall range of 0.3 to 14.6 mol m$^{-2}$ s$^{-1}$
(Table 1). The highest soil respiration values were observed in the UT sites in summer
(July, August; Table 1), with intermediate (1~3 mol m$^{-2}$ s$^{-1}$) and low values (< 1
mol m$^{-2}$ s$^{-1}$) in the BT and BR sites, respectively. Generally, soil COS exchange
varied from release to increasing uptake with increasing $CO_2$ production in a non-linear
way (Fig. 6a). The normalized ratio of COS to $CO_2$ fluxes (SRU; Eq. 2) varied from -
1.92 to +1.85 with an average value of -0.37 $\pm$ 0.31, with negative values indicating



COS uptake linked to $CO_2$ emission. SRU values showed response to both soil
temperature (Fig. 6b) and soil moisture (Fig. 6c), although with relatively low $R^2$ values.
Respiration increased with temperature while COS uptake declined and at temperature
above about 25 °C SRU turned positive when both COS and $CO_2$ are emitted from the
soil. SRU exhibited inverse relationships with soil moisture, with positive values in dry
soil and increasingly negative values with increasing soil moisture (Fig. 6c). Based on
its combined temperature ($Ts$) and moisture ($\theta$) response, SRU could be forecasted by
the following algorithm, which explained 67% of the observed variations (Eq. 8):
$$SRU = 0.01\exp(0.17Ts) - 0.02\theta - 1.00, \ R^2 = 0.67 \qquad (8)$$
ANOVA analysis results indicated that SRU was not significantly different among
the three observation microsites (BR, BT, and UT; $P > 0.05$). Between the seasonal
campaigns, however, SRU values peaked in summer (0.53 ± 0.66) with highest
averaged soil temperature (29 °C) and was significantly higher than winter SRU (-1.44
± 0.59) when soil temperature was lowest (11 °C; $P < 0.05$), and with no significant
difference in SRU among the other campaigns ($P > 0.05$).

**4. Discussions**
**4.1 Heterogeneity in soil COS exchange**
The observed soil-atmosphere COS exchange rates observed in this study (both
mean and range; Fig. 1, Table 1) are consistent with values reported in a range of other
ecosystems (-1.4 to -4.9 pmol m$^{-2}$ s$^{-1}$; Steinbacher et al., 2004;Kitz et al., 2017;White
et al., 2010;Berkelhammer et al., 2014), but lower than -11.0 to -11.8 pmol m$^{-2}$ s$^{-1}$ in a
riparian and subtropical forests (Berkelhammer et al., 2014;Yi et al., 2007). Soil COS
emissions were also observed in summer and spring campaigns, with maximal COS
emission consistent with the values of +1.8 to +2.6 pmol m$^{-2}$ s$^{-1}$ observed in a riparian
and alpine forests (Berkelhammer et al., 2014), but significantly lower than reported in
the senescing agricultural ecosystem (~30 pmol m$^{-2}$s$^{-1}$; Maseyk et al., 2014).
The observed range in the soil-atmosphere exchange fluxes reflected significant
heterogeneity on both the spatial and the temporal scales. The spatial scale



heterogeneity clearly reflected the contrasting microsite conditions with lower
temperatures and higher moisture under the trees (UT sites), compared to the higher
temperatures and lower moisture in exposed soil between rows (BR sites), with
intermediate, partially shaded, conditions between trees (BT sites). Indeed, a large
fraction of the variations in the COS flux (~75%) could be explained by a simple
algorithm as a function of these two variables, temperature and moisture. Note that
while temperature and $\theta$ co-varied in general, with high temperatures associated with
drier soil, under the wet UT conditions, sensitivity to temperature was significantly
reduced. In the dry soil conditions, emission was associated with high temperature, and
in the BR sites also with high solar radiation. However, all measurements we made in
dark chamber and could not involve photochemical production (Kitz et al., 2017).
Apparently even under dark conditions, high temperature can induced high emission
rates, as also noted when the thermal insolation on the soil chamber in the BR site was
incidentally removed and a large spike in temperature (52 $^{o}$C) and emission of 11.4
pmol m$^{-2}$ s$^{-1}$ was observed.
Temporal variations were observed both on the daily and seasonal time scales.
Diurnal changes were, however, minor compared to the changes from winter to summer
in all microsites. Shifts from uptake to emission were observed essentially only on the
seasonal time scale (Fig. 1). This likely reflected the dominance of soil moisture on the
COS flux rates. This is because $\theta$ did not change significantly on the daily scale, while
it changed significantly on across seasons (between 10.0 and 35.5% overall).
Temperatures did change over the daily cycle (e.g. 26.0 to 42.4 $^{o}$C in the BR site during
summer), although such changes are still smaller compared with the seasonal changes
in soil temperature (e.g. 10.5 to 31.8 $^{o}$C in the BR site). A dominant role of soil moisture
in explaining the variations in COS uptake is consistent with the results of Van Diest
and Kesselmeier (2008), but not with the negligible $\theta$ effects in grassland under
simulated drought (Kitz et al., 2017). Soil moisture can influence soil COS exchanges
by influencing CA enzymatic activity (Davidson and Janssens, 2006;Seibt et al., 2006),
changing soil gas diffusion rates (Ogée et al., 2016;Sun et al., 2015), and vegetation
root distribution and the effects of CA activity within plant roots (Seibt et al.,



2006;Viktor and Cramer, 2005;Whelan and Rhew, 2015). In this study, most of the roots
were distributed around the restricted trees' drip irrigation zone at UT sites, and was
sparse in the dryer areas, such as BR and BT sites (un-quantified observations).
At least part of the variations in soil COS fluxes could also reflect the differential
effects of environmental conditions on COS uptake and production process (Ogée et al.,
2016). COS uptake is thought to be related to carbonic anhydrase activity in soil
microorganisms (Piazzetta et al., 2015), such as Bacteria (Kamezaki et al., 2016;Kato
et al., 2008), or fungi (Bunk et al., 2017;Li et al., 2010;Masaki et al., 2016). Solubility
in soil water (with COS solubility of 0.8 ml ml$^{-1}$; Svoronos and Bruno, 2002) could also
be significant, especially in the UT microsites, influenced by the drip irrigation from
May to September that could involve water percolation to deeper soil layers. The drivers
of soil COS production are still unclear. COS could be produced by chemical processes
in the lab (Ferm, 1957), but can also be produced by biotic process in soils such as by
hydrolysis of metallic thiocyanates (Katayama et al., 1992) with thiocyanate hydrolase
(Conrad, 1996;Svoronos and Bruno, 2002) and hydrolysis of $CS_2$ (Cox et al.,
2013;Smith and Kelly, 1988). Fungi are also reported to be the source of COS (Masaki
et al., 2016). Additionally, abiotic thermal degradation of organic matter leading to COS
production maybe supported by the temperature sensitivity of COS emission in the BR
microsite where biotic processes can be expected to be minimized. Similar high
temperature-dependent soil COS emissions were reported in midlatitude forest
(Commane et al., 2015) and agricultural field (Maseyk et al., 2014). Lab incubation
results also indicated soil thermo production of COS with increasing temperature (Liu
et al., 2010;Whelan et al., 2016;Whelan and Rhew, 2015). Photochemical production
of soil COS was also proposed (Sun et al., 2015;Whelan and Rhew, 2015), and assumed
to be driven by ultraviolet fraction of incoming solar radiation (Kitz et al., 2017). Note,
however, that all measurements in the present study were made in the dark. In addition,
the chemical reaction of CO and $MgSO_4$ under heating could also produce COS (Ferm,
1957). Note that $MgSO_4$ has been reported in our study soil (Singer, 2007), and we
observed relatively high CO concentration in our field site.


## 4.2 Soil relative uptake

For COS application as a tracer of ecosystem $CO_2$ exchange quantifying the relationships between COS and $CO_2$ fluxes is important. This is done by assessing the 'relative uptake' (RU) of the $COS/CO_2$ flux rate ratio, normalized by the ambient atmospheric concentrations (that differ for the two gases by a factor of about $10^6$), as done at the leaf, (LRU) or ecosystem (ERU; e,g, Asaf et al., 2013). It was similarly applied to soil as SRU (Eq. 2; Berkelhammer et al., 2014). We use SRU values also to assess the relative importance of the soil COS flux compared with the canopy.

On average, the absolute value of SRU at our site was smaller than reported for riparian or pine forests (0.37 vs 0.76 and 1.08; Berkelhammer et al., 2014). This may reflect the contribution of COS emissions at BR and BT in summer, that were not observed in the forest study. Overall, the mean SRU values observed here indicated that the soil COS uptake flux was proportionally less than 40% of the soil respiration flux. In contrast with the canopy fluxes where the COS uptake flux is, proportionally, nearly twice as large as the $CO_2$ assimilation flux (LRU~1.7; see review of Whelan et al., 2018). In contrast to leaves with robust LRU value that tend toward a constant, SRU at our site varied between -1.92 and +1.85. However, this range was observed only in the dryer and exposed BR sites, while in the shaded and moist UT sites, it was much narrower, -0.13 to -0.79. Furthermore, it seems that the high SRU values (both positive and negative) represented conditions where the actual fluxes were small (COS uptake was on average -3.0 in the UT but only 0.1 pmol m$^{-2}$ s$^{-1}$ in the BR sites. It seems that the large SRU values in the BR microsites, were also associated with low soil respiration, 0.5    mol m$^{-2}$ s$^{-1}$ in BR sites, compared to 10 mmol m$^{-2}$ s$^{-1}$ in the UT sites. It is therefore possible that the low SRU values are the more significant for ecosystem scale studies and indicate a much smaller contribution to overall ecosystem fluxes than that of the canopy (SRU~0.4 vs LRU~1.7).

Differential effects of changing environmental conditions on production and uptake processes were reflected in relatively large spatial and temporal heterogeneity observed in the soil COS exchange at our site. However, the contrasting effects of production and emission may explain both the sharp increase in SRU values at high



temperatures as the effects of production counteract uptake (Fig. 6b), and the much
lower sensitivity to temperature of COS flux compared to that of $CO_2$ (Fig. 6a). Such
contrasting consumption/production effects may, in fact, reduce the magnitude of the
net flux of soil COS, and may explain the relatively narrow range of SRU values.

### 4.3 Soil COS profiles

Complementing our chamber measurements with soil profile measurements of

COS and $CO_2$ concentrations provided constrain on the relatively new surface soil COS
measurements and provided additional information on the possible location of the
source/sink in the soil. Using the near surface gradient yielded flux estimates
comparable to chamber measurements, providing a useful and rare quantitative
validation. For example, in May, the chamber and profile measurements were made at
about the same time (5th~9th May for chamber and 10th May for profile) and the
differences between chamber (all microsites) and gradient flux estimates, was
negligible (~0.2-0.6 pmol m$^{-2}$ s$^{-1}$). However, the profile results indicated in addition that
the sink/source activities concentrated at top soil layers, probably at around 5 cm depth,
as reflected in the minimum or maximum in gas concentrations (indicating also the need
for high vertical resolution in employing the profile approach). The variable profiles
observed below these points must reflect temporal dynamics in the sink/source
activities across the profile. The near surface peak activity makes it particularly
sensitive to variations in temperature and moisture, as indeed observed (Figs. 2, 3). Low
COS concentration in the lower parts of the profile may result from continuous removal
of soil COS and may indicate distribution of CA activity beyond the litter layer and the
soil surface (Seibt et al., 2006). COS production, however, seems to occur only near the
soil surface with no indication for production in deeper layer, consistent with its high
temperature sensitivity, and possibly also radiation (e.g. Kitz et al., 2017).

Note that the gradient method based on Fick's diffusion law have its own

limitations (Kowalski and Sánchezcañete, 2010;Sánchez-Cañete et al., 2017;Bekele et
al., 2007). However, it is simple low-cost approach and can help diagnose the
magnitude of soil fluxes, which can also help in identifying below ground processes



and their locations.

**5.  Conclusions**

Our detail analysis of the spatial and temporal variations in soil-atmosphere
exchange of COS provided new information on a key uncertainty in the application of
ecosystem COS flux to assess productivity. Furthermore, we provide constraint, and
validation of the close chamber measurements that are generally in use, by the
additional gradient approach. Our results show that both microsites and seasonal
variations in COS fluxes were related to soil moisture, temperature, and the distance
from the tree (likely reflecting root distribution), but we suggest that soil moisture is
the predominant environmental control over soil COS exchanges at our site. A simple
algorithm was sufficient to forecast most of the variations in soil COS flux supporting
its incorporated into ecosystem scale applications, as we recently demonstrated in a
parallel study at the same site (Yang et al., 2018).
Clearly, uncertainties are still associated with soil processes involving COS, the
differential effects of soil moisture, temperature, and communities of microorganisms
and are likely to contribute to both the spatial and temporal variations in soil net COS
exchange and require further research.

**Author contributions:**

DY designed the study; FY, RQ, FT, RS and DY performed the experiments. FY
and FT analysed the data. DY and FY wrote the paper with discussions and
contributions to interpretations of the results from all co-authors.

**Acknowledgements**

We are grateful to Omri Garini, Madi Amer, and Boaz Ninyo-Setter for their help. This work
was supported by the Minerva foundation, a joint NSFC-ISF grant 2579/16; Israel Science
Foundation (ISF 1976/17), the German Research Foundation (DFG) as part of the CliFF
Project, and the JNF-KKL. FY is supported by the National Natural Science Foundation of
China (41775105), and the Natural Science Foundation of Gansu Province (17JR5RA341).



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





**Figure captions:**

**Figure 1.** Spatial variability of soil COS flux at three sites, between trees (a), between rows (b), and under tree (c). Each figure shows the diurnal cycling of soil COS flux in the four campaigns. Each data point was the hourly mean ± 1 S.E. (N=3).

**Figure 2.** Relationship of soil COS flux with soil moisture. Soil fluxes and mositure data point represent the diurnal average (N=24) of each microsite and season (i.e. each measurement campaign). Error bars represent ± 1 S.E. around the mean; errors for flux are about the size of the symbols.

**Figure 3.** Soil COS fluxes response to soil temperature linearly. Soil fluxes and represent the diurnal average (N=24) of each site and season (campaign), mositure data represent one sampling per day. Error bars represent ± 1 S.E. around the mean. The marked (black) data point were collected during irrigation campaign (enhanced uptake) and were excluded from the regression.

**Figure 4.** Mean COS and $CO_2$ concentrations at different soil depth. The COS concentration decreases exponentially with soil depth. The data point is the mean of the combined data at each of the four measurement campaigns (N=4; ± 1 S.E.).

**Figure 5.** Soil COS and $CO_2$ concentration profiles at the three microsites in four measurement campaigns**.** The data points are the mean of all measurements in a campaign (N=4, ± 1 S.E.)

**Figure 6.** The relationships between soil COS and $CO_2$ flux rates (chamber measurements; a). The response of soil relative uptake (SRU; normalized ratio of COS to $CO_2$ fluxes) to soil temperature (b) and to soil water content (c). The data points represent the diurnal average (N=24) of each site and season (measurement campaign). Error bars represent ± 1 S.E. around the mean (often the size of the symbol).



**Table 1.** Mean values of soil COS and $CO_2$ flux rates across sites (BR, between rows;
BT, between trees; UT, under tree), and seasons, together with the normalized ratio of
COS/$CO_2$ fluxes (SRU), and the mean soil temperature at 5 cm depth ($Ts$) and soil water
content (% by wt; $\theta$).

| Campaigns | Sites | COS flux (pmol m$^{-2}$ s$^{-1}$) | $CO_2$ flux (μmol m$^{-2}$ s$^{-1}$) | SRU | $Ts$ (ºC) | $\theta$ (%) |
|---|---|---|---|---|---|---|
| August, 2015 | BR | 1.83±0.08 | 0.77±0.04 | 1.85 | 31.66±1.01 | 9.98±0.28 |
| | BT | 0.06±0.05 | 3.33±0.05 | 0.01 | 29.09±0.20 | 19.77±0.02 |
| | UT | -3.64±0.13 | 10.79±0.12 | -0.26 | 28.80±0.26 | 24.03±0.40 |
| | | | | | | |
| December, 2015 | BR | -0.74±0.07 | 0.30±0.02 | -1.92 | 10.50±0.17 | 23.33±1.89 |
| | BT | -2.52±0.10 | 1.21±0.03 | -1.62 | 11.20±0.19 | 24.22±0.94 |
| | UT | -3.87±0.08 | 3.81±0.07 | -0.79 | 12.17±0.16 | 26.11±1.01 |
| | | | | | | |
| May, 2016 | BR | -0.77±0.02 | 0.32±0.02 | -1.88 | 21.67±0.32 | 15.56±0.38 |
| | BT | -0.05±0.04 | 1.31±0.05 | -0.03 | 22.20±0.34 | 15.70±1.03 |
| | UT | -1.80±0.11 | 10.78±0.54 | -0.13 | 20.35±0.38 | 22.11±1.44 |
| | | | | | | |
| July, 2016 | BR | 0.21±0.04 | 0.79±0.05 | 0.21 | 29.66±0.60 | 14.73±0.57 |
| | BT | 0.76±0.09 | 1.97±0.04 | 0.30 | 26.68±0.15 | 17.49±0.70 |
| | UT | -2.67±0.09 | 14.58±0.40 | -0.14 | 27.83±0.34 | 35.47±3.47 |






**Table 2.** Estimates of soil COS and $CO_2$ fluxes from soil concentration gradient measurements ($Ts$, soil temperature; $\theta$, soil water content; BR, between rows; BT, between trees; UT, under tree.)

| Campaigns | Sites | COS flux (pmol m⁻² s⁻¹) | $CO_2$ flux (μmol m⁻² s⁻¹) | $CO_2$ diffusion coefficient (mm² s⁻¹) | COS diffusion coefficient (mm² s⁻¹) | $Ts$ (°C) | $\theta$ (%) |
|---|---|---|---|---|---|---|---|
| March, 2016 | BR | -1.31 | 2.34 | 5.21 | 4.40 | 17.9 | 19.4 |
|  | BT | -1.15 | 2.21 | 4.80 | 4.05 | 16.2 | 21.8 |
|  | UT | -2.10 | 5.89 | 4.76 | 4.02 | 17.3 | 22.4 |
| April, 2016 | BR | -1.55 | 1.07 | 6.66 | 5.62 | 23.0 | 11.0 |
|  | BT | -0.89 | 1.14 | 6.44 | 5.43 | 20.4 | 11.6 |
|  | UT | -1.74 | 4.73 | 6.01 | 5.07 | 22.4 | 15.2 |
| May, 2016 | BR | -0.98 | 2.21 | 5.68 | 4.79 | 21.9 | 17.4 |
|  | BT | -0.51 | 1.24 | 5.06 | 4.27 | 22.0 | 21.6 |
|  | UT | -1.20 | 11.36 | 3.11 | 2.63 | 20.1 | 34.5 |
| June, 2016 | BR | 1.55 | 2.63 | 6.61 | 5.57 | 35.9 | 15.5 |
|  | BT | -1.17 | 2.60 | 5.20 | 4.39 | 26.3 | 21.7 |
|  | UT | -1.19 | 11.85 | 3.02 | 2.55 | 22.9 | 35.6 |








**Figure 1**

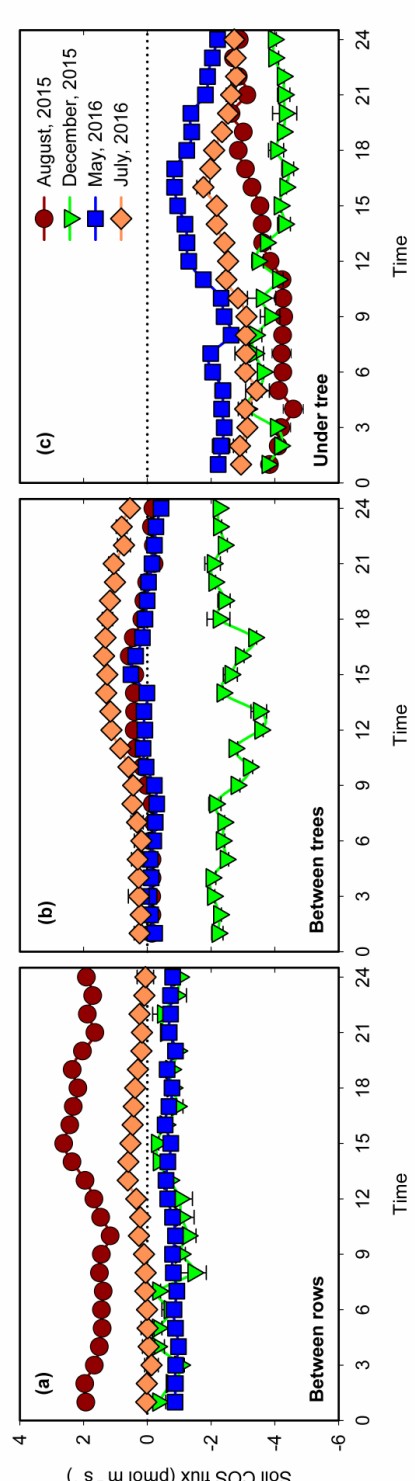

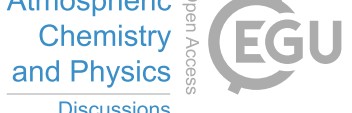



**Figure 2**

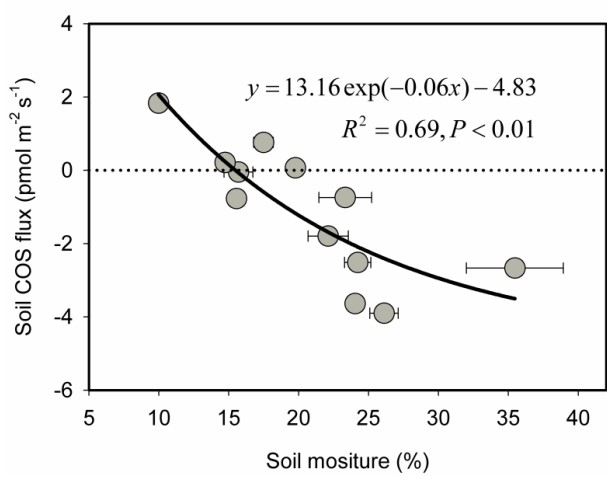





**Figure 3**

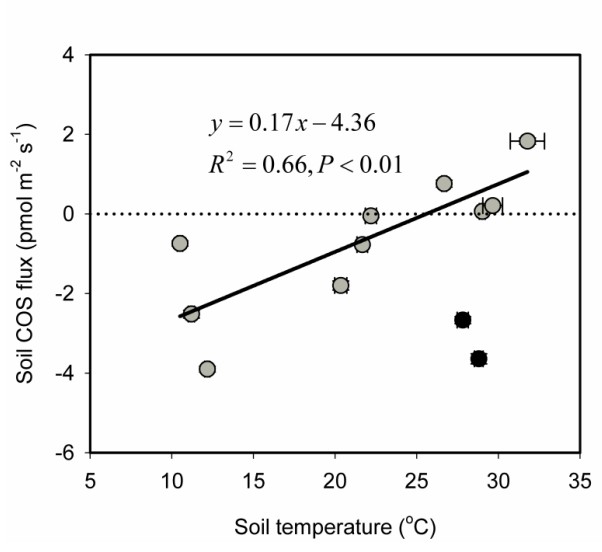





**Figure 4**

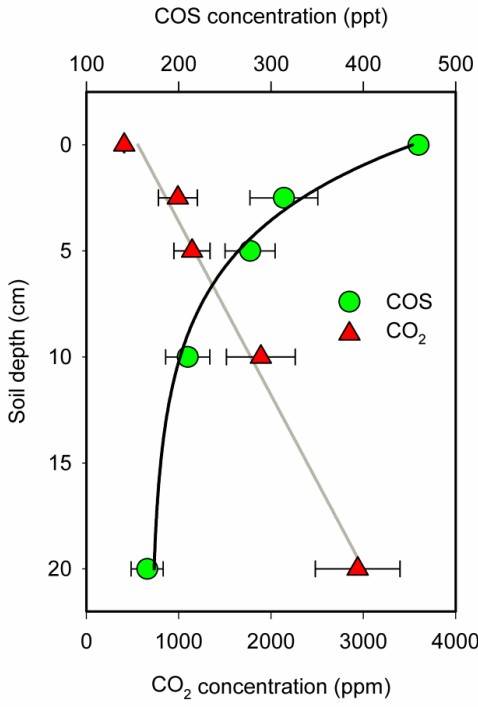




**Figure 5**

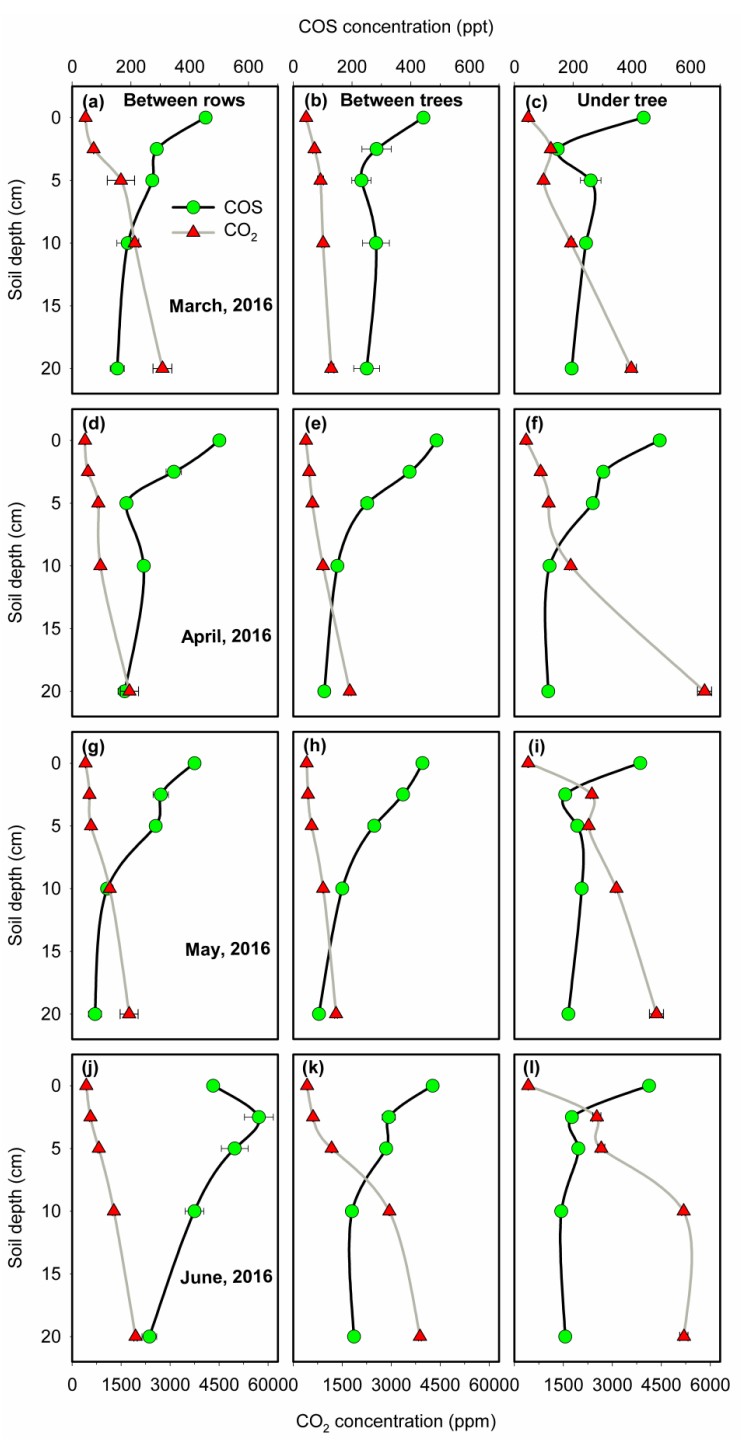



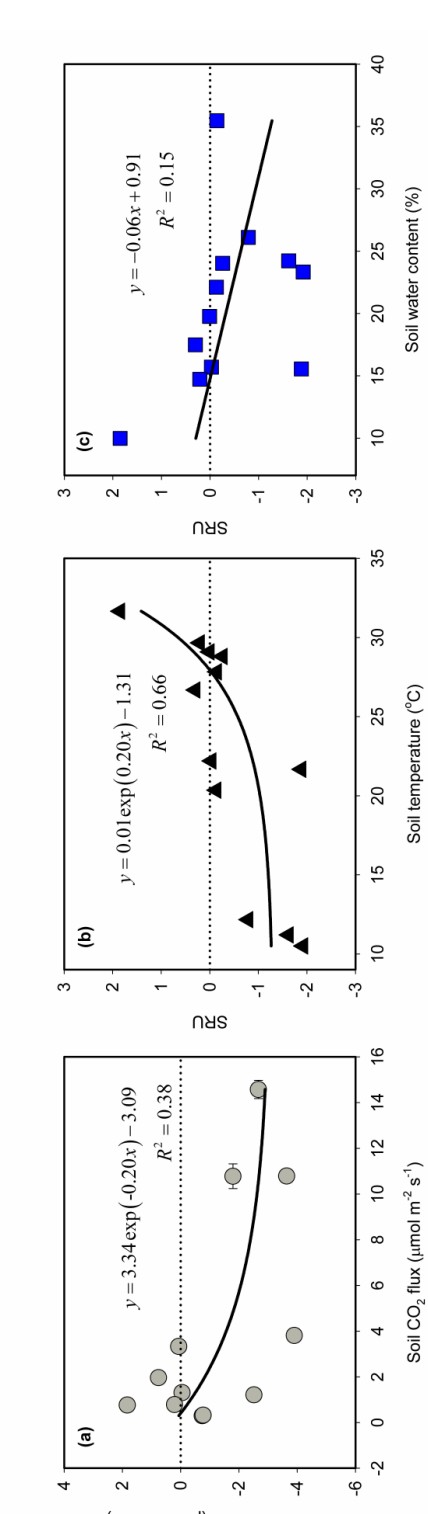

**Figure 6**