# Peer review of "Soil-atmosphere exchange of carbonyl sulfide in Mediterranean"

_Atmospheric Chemistry and Physics, 2018_

## Referee Comment (RC1) · Anonymous Referee #1 · 19 Dec 2018

In this study, the authors present soil fluxes of COS from a Mediterranean soil and examined the spatial and temporal variation of those fluxes. The authors find substantial differences in space and time, from emissions to uptake of COS, of which the variability is predominantly driven by soil moisture. Although the English grammar can be improved, this is a well-structured paper with a good discussion of the results. This study is a nice contribution to understanding the COS exchange at different ecosystems, of which the Mediterranean soil was still missing. Below are general and specific comments:

General comments

In the discussion on the different COS soil fluxes at the different sites it would be very helpful to visualise in Fig. 2, 3 and 6 to which site each data point belongs (e.g. use

a color per site). This would make it easier to understand how soil moisture varied between the different sites, instead of referring to the numbers in Table 1. Also, the result sections 3.1 and 3.2 could be better readable if the data are discussed while referring to the figures, instead of referring to Table 1. If the data points from the different sites are visualised in the figures I think that Table 1 would be unnecessary (could be an appendix?), as all the information is then presented in the figures.

The discussion focusses a lot on the variability of SRU, and I miss some statements on the importance and application of SRU earlier in the text. Moreover, the difference in size between SRU and LRU is used to discuss the implications of the soil COS fluxes for the use of COS as tracer for GPP, which I do not find very intuitive. Instead, I would compare the size of soil COS fluxes directly with that of ecosystem scale COS exchange, not that of the ratios of COS to CO2 fluxes. Comparing the size of soil COS exchange with ecosystem COS exchange would also make a very nice link with the results from the parallel study (Yang et al., 2018, Glob. Change Biol.), which has now only been little discussed.

It would also strengthen the message of the paper if the variability of the soil COS uptake in space and time is discussed in light of using COS as tracer for GPP. The soil COS exchange is relatively small compared to the ecosystem COS exchange, but the variability in space and time seems significant and may still complicate the COS tracer method? It would be good to discuss the implications of the variability of the soil COS exchange on the COS tracer method.

Specific comments

Abstract

P2L19: what do you mean with "exposed"? Exposed to the sun?

P2L20: "weighted mean uptake values"? Do you mean simply the average? Or what is it weighted by?

Introduction:

P3L60: "In spite of these efforts, more field measurements of soil COS exchange are clearly needed". It would be good to mention here that also contrasting ecosystems need to be studied. Previous studies have focused on agricultural soils (Maseyk et al., 2014), wetlands (Whelan et al., 2013), boreal forest soils (Sun et al., 2018), grasslands (Kitz et al., 2017), but several ecosystems are understudied, and the Mediterranean soil is a highly needed addition to this.

I like that the soil profiles of COS have now also been measured (for the first time?). It can be mentioned that the soil profile measurements will also be useful for validation of soil models of COS exchange (Sun et al., 2015.).

Materials and methods:

One of the first questions I got while reading the manuscript was what the role is of photoproduction on the soil COS emissions, and only later I read that dark soil chambers were used. It would be good to mention already in the methods section that the chambers were dark and that photoproduction is not expected to play a role in the results.

It was shown by Kooijmans et al. (2016, AMT), that measurements of COS made with the QCL can be biased at high H2O. Were the measurements corrected for the effect of water vapour?

If available, it would be good to add more soil characteristics for this site, like pH and soil porosity.

Results:

P6L146: What do you mean with "interactions between microsite and season"?

P7L 178-179: "The response of soil COS fluxes to soil temperature varied among the three measurement sites (Fig. 3)". This can only be judged from Fig. 3 when the data

points in Fig. 3 get marked by site.

Discussions:

P11L289: "Temperatures did change over the daily cycle...". Soil temperature?

P13L334-335: "We use SRU values also to assess the relative importance of the soil COS flux compared with the canopy.". I don't find this approach very intuitive. Why not simply compare the size of the soil COS flux with that of the ecosystem COS flux?

P13L336: "... the absolute value of SRU...". The sign of SRU shifts from negative to positive due to change in sign of the COS flux. So I would say that referring to the absolute value is not appropriate here.

P13L337: Reference to the pine forest soil (Sun et al., 2018) is missing.

P13L342: A reference to Whelan et al. 2018 is given with LRU $\sim$ 1.7, which is the average over a large range of plant species. Why not refer to Yang et al. 2018 with LRU = 1.6 (at high light) that is specific for this site?

Conclusions:

P15L393: "we provide constraint, and validation of the closed chamber measurements ... by the additional gradient approach". What do you mean with "constraint"? I would just call it validation.

P15L397: This hypothesis on root distribution is introduced only very late in the manuscript, and it would be good to describe the different root distribution for the different sites already in the methods section.

Tables and figures:

Information in Table 1 is presented in Figures 2, 3 and 6, and the results section 3.1 and 3.2 could be better readable if it refers to the figures rather than the table.

Fig. 2, 3 and 6 would benefit from having the data points marked by site.

Figure 4 does not provide more information than Fig. 5, and I would consider removing it.

Technical corrections

-Be consistent throughout the text in the sign convention of fluxes. E.g. uptake of +2.5 pmol m-2 s-1, or a flux of -2.5 pmol m-2 s-1.

-Be consistent with the number of digits of fluxes. E.g. P9L222: -1.0 +/- 0.26 pmol m-2 s-1, which is -1.02 +/- 0.26 pmol m-2 s-1 in the abstract.

-CO2 flux units are missing the micro sign.

-mositure = moisture (both in text and figure axis labels).

Some textual corrections:

P3L42: "non-leaf contributions to the net ecosystem COS flux".

P3L50: Event = Even

P4L83: "described by Asaf et al. (2013).

P5L109: "...from the normalized ratio of CO2 respiration to COS uptake (negative values) or emission (positive values) fluxes".

P6L125: "soil COS and CO2 fluxes were estimated based on Fick's first law: ..."

P6 Eq4: Ts is not defined in the text, but later T is defined. If Ts and T are the same then make it consistent throughout the text.

P6L146: "both the spatial (microsites) and temporal (seasonal) scale".

P7L157: "In the UT site..."

P7L176: "The fit to the data..."

P8L206: Reference to Fig. 4 should be Fig. 5?

[Figure]

P11L279: induced = induce.

P11L288: remove "on".

P12L315: maybe = may be

P12L319: Lab incubation results also indicated thermal production of COS in soil. . ."

P15L391: "Our detailed analysis..."

P21L569-571: caption of Figure 3, this sentence doesn't flow logically.

---

## Referee Comment (RC2) · Whelan (Referee) · 8 Jan 2019

Dear Dr. Yang and others,

This study presents a good, field-based approach to characterizing OCS soil fluxes. Observations of OCS in the soil profile will do well to inform the process-based OCS soil modeling that has been framed in the past (Ogée et al., 2015; Sun et al., 2015).

At the same time, I have two methodological concerns about the profile measurements (L118-123). One is that decabon tubing might be unsuitable for "static" measurements. In other words, you might get some production or absorption from the tubing itself when put in contact with an air sample for long periods. For applications where the tubing is flushed, it appears to be fine. Within our own QCL, we replaced the decabon tubing

for static applications, where an air sample had to be contained within the sample cell and associated tubing for several minutes. Roisin Commane also reported similar issues. One possibility is that the decabon tubing you used was of a different age or a different formulation of plastic and doesn't have these problems. Reporting a simple blank experiment, which you've probably already performed, would clear things up. Perhaps trapping a sample of known OCS concentration in a few coils of tubing, waiting a day, then measuring it again, will show whatever uncertainty might be introduced by using this material.

The second concern I have is one about drawing air samples out of a soil profile. Unless a small volume (e.g. a stainless steel cup with a screen on the bottom) is buried at each depth, there is not that much volume of air to draw out. The methods suggest 400 mL/min for 5 min at each depth, with repeated measurements every 40 minutes for 5 total measurements at each depth. It's difficult to tell what is actually being sampled here: the soil depths start at 2.5 cm. If we think of the air as being drawn from a sphere around the sampling point, it seems like each 2L sample will end up drawing ambient air from the above-soil atmosphere. So, essentially you end up looking at ambient air that has been very recently drawn through different soil volumes. Is this impression accurate?

The introduction needs some touching up to be more useful for the reader. There is some vagueness with the wording and the literature reviewed could be better situated. The non-leaf fluxes are not "unknown" (L42) but poorly characterized. The word "significant" (L43, L47, L160, L170) is vague unless a level of significance is defined, as in L184 and L246.

The narrative presented regarding the changing ideas around OCS soil fluxes starting on L45 is slightly misleading: "exchanges were often considered small compared to plant uptake". The Whelan (2013) paper cited there showed a huge OCS emission from wetland soils that far exceeded any plant uptake. Most of the first work on OCS exchange was from wetlands, where large emissions were considered worthy of study

and relatively easier to measure. At any rate, I'm providing a perspective here to help clarify your own structure, without expecting to include all or even any of this in your introduction. There are many studies that have furthered our understanding of OCS soil exchange. For your agricultural study site, you could focus the discussion on the curious problem of high OCS production in agriculturally managed soils. The progression of our understanding of soil OCS exchange moves forward like this:

(1) At first, wetlands are considered a huge source of OCS, though most of the data collected from pre-1992 have methodological problems detailed in (Castro and Galloway, 1991). OCS production is linked to redox potential. Our set of valid wetlands measurements to date are summarized in Whelan et al. (2018) Figure 4.

(2) Kesselmeier et al. (1999) performs the first thorough set of measurements on a single, oxic, agricultural soil and finds only OCS uptake with a soil moisture and temperature optimum. Adding a carbonic anhydrase inhibitor reduced uptake. This study produced a model of soil OCS exchange with a maximum of 10 pmol/m2/sec and suggested that oxic soils are entirely a sink, probably biotic. This idea was revisited with 4 soils in Van Diest and Kesselmeier (2008).

(3) Watts (2000) is a review paper that clarified the thinking of the time: anoxic soils are a large OCS source and oxic soils are a small sink.

(4) This view was challenged in the literature when Maseyk et al. (2014) dragged a QCL laser into a wheat field in Oklahoma. They installed an automated soil chamber which enabled near continuous observations of oxic soil OCS fluxes in the field. I think the Liu paper was a lab-based experiment, and not enough of a big deal was made over how surprising it was for dried rice paddy soil to produce OCS.

(5) The authors of the Maseyk study sent me a soil sample from the Oklahoma site, which I did a series of laboratory incubations with. I used a GC/MS, the more traditional measurement approach, to confirm that the Maseyk findings weren't the result of some sort of methodological problem. We sterilized the soil and subjected it to full spectrum

light, and demonstrated that OCS production was linked to both light and temperature and was probably abiotic (Whelan and Rhew, 2015).

(6) We then got soil samples from many, many sites and repeated the approach with a QCL in Whelan et al. (2016). We showed that air-dried soils in all biomes except deserts exhibit net OCS production, increasing exponentially with temperature. If one subtracts this curve from measurements made at other soil moistures, a curve resembling the initial Kesselmeier (1999) model can be recovered. This showed that abiotic OCS production and biotic consumption were occurring, probably in most soils. No one knew at this point why agricultural soils had such large magnitude emissions with temperature.

(7) Kitz et al., (2017) was the first field study to make conclusions about the effect of light in the field. It should be noted that this was an agriculturally managed soil, not a natural grassland. This is something to keep in mind in your discussion of it (L293/4).

(8) Finally, Kaisermann (2018) showed that nitrogen suppresses OCS consumption, providing a path to solve the mystery of why some agricultural soils appear to have such large OCS production compared to other oxic soils.

The discussion on L283 to L326 could be improved, too. In particular, the role of carbonic anhydrase is introduced multiple times. Also, I invite you to revisit the solubility/hydrolysis figures in Whelan et al., (2018) and calculate the proportion of OCS that might be dissolved then lost to hydrolysis considering your soil moisture content and OCS within-soil concentrations. The mention of the CO and $MgSO_4$ is an interesting idea that I wish we could explore more. While I know that your QCL is capable measuring CO, the ambient standards are tricky to maintain. Was CO measured? Is it possible to pull out any interesting relationships with CO concentration and temperature? Related, were the water vapor measurements mentioned in L79 calibrated? If so, how? It doesn't look like you used these, except indirectly for a water vapor correction on the other two gases.

Section 4.2 focuses on SRU values. Often the point of understanding soil OCS exchange is to "correct" ecosystem level exchange so that a more accurate GPP estimate can be made. A possible motivation for comparing soil OCS exchange and soil CO2 fluxes would be to try and estimate OCS soil fluxes from soil respiration model output. Otherwise, I'm not clear on what we gain by using SRU. In short, more justification is needed for this section.

Overall, this is an important and interesting study. Your ability to predict most OCS soil fluxes using only two variables is a great achievement and also heartening for other modeling efforts. Soil profiles would be very useful to support process-based soil OCS exchange modeling. I look forward to seeing this paper in its final form.

Sincerely,

Mary Whelan

References Castro, M. S. and Galloway, J. N.: A comparison of sulfur-free and ambient air enclosure techniques for measuring the exchange of reduced sulfur gases between soils and the atmosphere, J. Geophys. Res., 96(D8), 15427–15, 1991.

Kaisermann, A., Jones, S., Wohl, S., Ogée, J. and Wingate, L.: Nitrogen Fertilization Reduces the Capacity of Soils to Take up Atmospheric Carbonyl Sulphide, Soil Systems, 2(4), 62, 2018.

Kesselmeier, J., Teusch, N. and Kuhn, U.: Controlling variables for the uptake of atmospheric carbonyl sulfide by soil, J. Geophys. Res., 104(D9), 11577–11584, 1999.

Kitz, F., Gerdel, K., Hammerle, A., Laterza, T., Spielmann, F. M. and Wohlfahrt, G.: In situ soil COS exchange of a temperate mountain grassland under simulated drought, Oecologia, 183(3), 851–860, 2017.

Maseyk, K., Berry, J. A., Billesbach, D., Campbell, J. E., Torn, M. S., Zahniser, M. and Seibt, U.: Sources and sinks of carbonyl sulfide in an agricultural field in the Southern Great Plains, Proc. Natl. Acad. Sci. U. S. A., 111(25), 9064–9069, 2014.

[Figure]

Ogée, J., Sauze, J., Kesselmeier, J., Genty, B., Van Diest, H., Launois, T. and Wingate, L.: A new mechanistic framework to predict OCS fluxes from soils, Biogeosci. Discuss., 12(18), 15687–15736, 2015. Sun, W., Maseyk, K., Lett, C. and Seibt, U.: A soil diffusion–reaction model for surface COS flux: COSSM v1, Geosci. Model Dev., 8(10), 3055–3070, 2015.

Van Diest, H. and Kesselmeier, J.: Soil atmosphere exchange of carbonyl sulfide (COS) regulated by diffusivity depending on water-filled pore space, Biogeosciences, 5(2), 475–483, 2008.

Watts, S. F.: The mass budgets of carbonyl sulfide, dimethyl sulfide, carbon disulfide and hydrogen sulfide, Atmos. Environ., 34(5), 761–779, 2000.

Whelan, M. and Rhew, R.: Carbonyl sulfide produced by abiotic thermal and photo-degradation of soil organic matter from wheat field substrate, J. Geophys. Res. Biogeosci., 2014JG002661, 2015.

Whelan, M. E., Min, D.-H. and Rhew, R. C.: Salt marshes as a source of atmospheric carbonyl sulfide, Atmos. Environ., 73, 131–137, 2013.

Whelan, M. E., Hilton, T. W., Berry, J. A., Berkelhammer, M., Desai, A. R. and Campbell, J. E.: Carbonyl sulfide exchange in soils for better estimates of ecosystem carbon uptake, Atmos. Chem. Phys., 16(6), 3711–3726, 2016.

Whelan, M. E., Lennartz, S. T., Gimeno, T. E., Wehr, R., Wohlfahrt, G., Wang, Y., Kooijmans, L. M. J., Hilton, T. W., Belviso, S., Peylin, P., Commane, R., Sun, W., Chen, H., Kuai, L., Mammarella, I., Maseyk, K., Berkelhammer, M., Li, K.-F., Yakir, D., Zumkehr, A., Katayama, Y., Ogée, J., Spielmann, F. M., Kitz, F., Rastogi, B., Kesselmeier, J., Marshall, J., Erkkilä, K.-M., Wingate, L., Meredith, L. K., He, W., Bunk, R., Launois, T., Vesala, T., Schmidt, J. A., Fichot, C. G., Seibt, U., Saleska, S., Saltzman, E. S., Montzka, S. A., Berry, J. A. and Campbell, J. E.: Reviews and syntheses: Carbonyl sulfide as a multi-scale tracer for carbon and water cycles, Biogeosciences, 15(12),

3625–3657, 2018.

---

## Author Comment (AC1) · 23 Feb 2019

We would like to thank the two reviewers for their helpful comments on this work. We fully addressed all the reviewer's comments, as described below. We believe the revisions improved the paper, while there is no change in any of the results or conclusions. The track corrected version of the manuscript is attached at the end.

**Referee #1**

In this study, the authors present soil fluxes of COS from a Mediterranean soil and examined the spatial and temporal variation of those fluxes. The authors find substantial differences in space and time, from emissions to uptake of COS, of which the variability is predominantly driven by soil moisture. Although the English grammar can be improved, this is a well-structured paper with a good discussion of the results. This study is a nice contribution to understanding the COS exchange at different ecosystems, of which the Mediterranean soil was still missing. Below are general and specific comments:

**Response:** Thanks for your kind comments.

General comments

1) In the discussion on the different COS soil fluxes at the different sites it would be very helpful to visualise in Fig. 2, 3 and 6 to which site each data point belongs (e.g. use a color per site). This would make it easier to understand how soil moisture varied between the different sites, instead of referring to the numbers in Table 1. Also, the result sections 3.1 and 3.2 could be better readable if the data are discussed while referring to the figures, instead of referring to Table 1. If the data points from the different sites are visualised in the figures I think that Table 1 would be unnecessary (could be an appendix?), as all the information is then presented in the figures.

**Response:** As suggested, we now both refer to the Figures, and marked the points by site. Since Table 1 contain additional data and is cited many times throughout other sections we also left Table 1 in the main text.

2) The discussion focusses a lot on the variability of SRU, and I miss some statements on the importance and application of SRU earlier in the text. Moreover, the difference in size between SRU and LRU is used to discuss the implications of the soil COS fluxes for the use of COS as tracer for GPP, which I do not find very intuitive. Instead, I would compare the size of soil COS fluxes directly with that of ecosystem scale COS exchange, not that of the ratios of COS to CO2 fluxes. Comparing the size of soil COS exchange with ecosystem COS exchange would also make a very nice link with the results from the parallel study (Yang et al., 2018, Glob. Change Biol.), which has now only been little discussed.

**Response:** Point is well taken. We added a full paragraph in the Introduction about SRU and a paragraph in the Discussion reintroducing the comparison of the soil and fluxes to canopy from Yang et al. 2018.

3) It would also strengthen the message of the paper if the variability of the soil COS uptake in space and time is discussed in light of using COS as tracer for GPP. The soil COS exchange is relatively small compared to the ecosystem COS exchange, but the variability in space and time seems significant and may still complicate the COS tracer method? It would be good to discuss the implications of the variability of the soil COS exchange on the COS tracer method.

**Response:** This is to a significant extent covered by in the previous comment above. We also note that this discussion is provided in detail in the Yang et al 2018 paper where both soil and canopy fluxes are reported, which we don't think should be repeated here. Instead, here we focus on providing more data that could help characterize and forecast the soil component, and validate it.

**Specific comments**

Abstract

P2L19: what do you mean with "exposed"? Exposed to the sun?

**Response:** Changed so 'sun-exposed'.

P2L20: "weighted mean uptake values"? Do you mean simply the average? Or what is it weighted by?

**Response:** corrected ('weighted' deleted).

Introduction:

P3L60: "In spite of these efforts, more field measurements of soil COS exchange are clearly needed". It would be good to mention here that also contrasting ecosystems need to be studied. Previous studies have focused on agricultural soils (Maseyk et al., 2014), wetlands (Whelan et al., 2013), boreal forest soils (Sun et al., 2018), grasslands (Kitz et al., 2017), but several ecosystems are understudied, and the Mediterranean soil is a highly needed addition to this.

I like that the soil profiles of COS have now also been measured (for the first time?). It can be mentioned that the soil profile measurements will also be useful for validation of soil models of COS exchange (Sun et al., 2015.).

**Response:** Thank you. The suggested text was added.

Materials and methods:

One of the first questions I got while reading the manuscript was what the role is of photoproduction on the soil COS emissions, and only later I read that dark soil chambers were used. It would be good to mention already in the methods section that the chambers were dark and that photoproduction is not expected to play a role in the results.

**Response:** Done.

It was shown by Kooijmans et al. (2016, AMT), that measurements of COS made with the QCL can be biased at high H2O. Were the measurements corrected for the effect of water vapour?

**Response:** Yes, correction was made. This is now indicated in the Method and the reference cited.

If available, it would be good to add more soil characteristics for this site, like pH and soil porosity.

**Response:** More information added (but we don't have soil porosity data).

Results:

P6L146: What do you mean with "interactions between microsite and season"?

**Response:** The confusing term was deleted.

P7L 178-179: "The response of soil COS fluxes to soil temperature varied among the three measurement sites (Fig. 3)". This can only be judged from Fig. 3 when the data points in Fig. 3 get marked by site.

**Response:** Marked as suggested.

Discussions:

P11L289: "Temperatures did change over the daily cycle. . .". Soil temperature?

**Response:** Changed to "Soil temperature".

P13L334-335: "We use SRU values also to assess the relative importance of the soil COS flux compared with the canopy.". I don't find this approach very intuitive. Why not simply compare the size of the soil COS flux with that of the ecosystem COS flux?

**Response:** See response to main comment 2.

P13L336: ". . . the absolute value of SRU. . .". The sign of SRU shifts from negative to positive due to change in sign of the COS flux. So I would say that referring to the absolute value is not appropriate here.

**Response**: Text improves as noted for main comments 2. Note that either positive or negative, a small SRU, below 1, is indicative of "suppressed" COS flux compared to $CO_2$, as opposed to LRU where the reverse is true.

P13L337: Reference to the pine forest soil (Sun et al., 2018) is missing.

**Response**: Added.

P13L342: A reference to Whelan et al. 2018 is given with LRU 1.7, which is the average over a large range of plant species. Why not refer to Yang et al. 2018 with LRU = 1.6 (at high light) that is specific for this site?

**Response:** Done.

Conclusions:

P15L393: "we provide constraint, and validation of the closed chamber measurements . . . by the additional gradient approach". What do you mean with "constraint"? I would just call it validation.

**Response:** Done.

P15L397: This hypothesis on root distribution is introduced only very late in the manuscript, and it would be good to describe the different root distribution for the different sites already in the methods section.

**Response:** In fact, root distribution was briefly discussed in the Discussion, section 4.1. We now added a comment on this also in the Methods. Our data are only on distance from the trees, so we could comment on this "hypothesis" as a logical possibility.

Tables and figures:

Information in Table 1 is presented in Figures 2, 3 and 6, and the results section 3.1 and 3.2 could be better readable if it refers to the figures rather than the table.

**Response:** Done.

Fig. 2, 3 and 6 would benefit from having the data points marked by site.

**Response:** Done.

Figure 4 does not provide more information than Fig. 5, and I would consider removing it.

**Response:** It is true the two figures refer to the same data, but we they are complementary: with Fig. 4 showing the "integrated" response with rather dominant patterns, which is the main message. But Fig. 5 shows that behind these patterns there are interesting variations that could be open for interoperations by the readers. Since these profile data are rather unique, we feel that showing the entire picture is justified.

Technical corrections

-Be consistent throughout the text in the sign convention of fluxes. E.g. uptake of +2.5 pmol m-2 s-1, or a flux of -2.5 pmol m-2 s-1.

**Response:** Done.

-Be consistent with the number of digits of fluxes. E.g. P9L222: -1.0 +/- 0.26 pmol m-2 s-1, which is -1.02 +/- 0.26 pmol m-2 s-1 in the abstract.

**Response:** Done.

-CO2 flux units are missing the micro sign.

**Response:** Done.

-mositure = moisture (both in text and figure axis labels).

**Response:** Done.

Some textual corrections:

P3L42: "non-leaf contributions to the net ecosystem COS flux".

**Response:** Done.

P3L50: Event = Even

**Response:** Done.

P4L83: "described by Asaf et al. (2013).

**Response:** Done.

P5L109: ". . .from the normalized ratio of CO2 respiration to COS uptake (negative values) or emission (positive values) fluxes".

**Response:** Done.

P6L125: "soil COS and CO2 fluxes were estimated based on Fick's first law: . . ."

**Response:** Done.

P6 Eq4: Ts is not defined in the text, but later T is defined. If Ts and T are the same then make it consistent throughout the text.

**Response:** Corrected.

P6L146: "both the spatial (microsites) and temporal (seasonal) scale".

**Response:** Corrected.

P7L157: "In the UT site. . ."

**Response:** Done.

P7L176: "The fit to the data. . ."

**Response:** Done.

P8L206: Reference to Fig. 4 should be Fig. 5?

**Response:** Done.

P11L279: induced = induce.

**Response:** Done.

P11L288: remove "on".

**Response:** Done.

P12L315: maybe = may be

**Response:** Done.

P12L319: Lab incubation results also indicated thermal production of COS in soil. . .”

**Response:** Done.

P15L391: "Our detailed analysis..."

**Response:** Done.

**P21L569-571: caption of Figure 3, this sentence doesn't flow logically.**

**Response:** Revised.
* * *
**Referee #2**

This study presents a good, field-based approach to characterizing OCS soil fluxes. Observations of OCS in the soil profile will do well to inform the process-based OCS soil modeling that has been framed in the past (Ogée et al., 2015; Sun et al., 2015).

**Response:** Thanks for the positive comments.

At the same time, I have two methodological concerns about the profile measurements (L118-123). One is that decabon tubing might be unsuitable for "static" measurements. In other words, you might get some production or absorption from the tubing itself when put in contact with an air sample for long periods. For applications where the tubing is flushed, it appears to be fine. Within our own QCL, we replaced the decabon tubing for static applications, where an air sample had to be contained within the sample cell and associated tubing for several minutes. Roisin Commane also reported similar issues. One possibility is that the decabon tubing you used was of a different age or a different formulation of plastic and doesn't have these problems.

Reporting a simple blank experiment, which you've probably already performed, would clear things up. Perhaps trapping a sample of known OCS concentration in a few coils of tubing, waiting a day, then measuring it again, will show whatever uncertainty might be introduced by using this material.

**Response:** As now clarified, samples were not static and were not kept inside closed-off tubing before analysis. Tubing were flushed for several turnovers of tubing volume before used for analysis. Note that even the longest 20 cm deep tubing had a volume of ca 10 ml requiring few sec flushing, and other connecting tubing were flushed with atmospheric air before soil sampling.

The second concern I have is one about drawing air samples out of a soil profile. Unless a small volume (e.g. a stainless steel cup with a screen on the bottom) is buried at each depth, there is not that much volume of air to draw out. The methods suggest 400 mL/min for 5 min at each depth, with repeated measurements every 40 minutes for 5 total measurements at each depth. It's difficult to tell what is actually being sampled here: the soil depths start at 2.5 cm. If we think of the air as being drawn from a sphere around the sampling point, it seems like each 2L sample will end up drawing ambient air from the above-soil atmosphere. So, essentially you end up looking at ambient air that has been very recently drawn through different soil volumes. Is this impression accurate?

**Response:** Thank you for noting the confusing part, and hopefully things are clearer now: The flow rate was in fact 80 ml/min for 5 min (total volume of 400 ml pumped), and data used only from the 3rd min window. For soil tube volumes of 3-12 ml, depending on depth, 2-minute flush was more than sufficient. The QCL sampling cell (500 ml) was setup at 15 torr and therefore another 1 min at 80 ml/min sufficiently flush the cell (~8 turnovers). The main cause of atmospheric leaks in preliminary trials was leaks around the outside of the tubes (but fixed). In addition, the five sampling cycles per site were never consecutive and were, in fact, hours apart (sometime the following day). And the soil tubes to different depths were not in bundles, but rather spread to avoid communication between sampling points during sampling. (There is no way that we could consistently observed thousands of ppm of $CO_2$ above ambient, and hundreds of ppt of COS below ambient a few cm below the surface if the system was leaking from the atmosphere).

The introduction needs some touching up to be more useful for the reader. There is some vagueness with the wording and the literature reviewed could be better situated. The non-leaf fluxes are not "unknown" (L42) but poorly characterized.
**Response:** Changed as suggested.

The word "significant" (L43, L47, L160, L170) is vague unless a level of significance is defined, as in L184 and L246.
**Response:** "significant" was replaced.

The narrative presented regarding the changing ideas around OCS soil fluxes starting on L45 is slightly misleading: "exchanges were often considered small compared to plant uptake". The Whelan (2013) paper cited there showed a huge OCS emission from wetland soils that far exceeded any plant uptake. Most of the first work on OCS exchange was from wetlands, where large emissions were considered worthy of study and relatively easier to measure.
**Response:** Indeed, we contrasted references on small and large soil fluxes, noting wetlands. But we nevertheless revised the language as suggested to avoid any 'misleading' (e.g. changed to, "in some cases small fluxes….", and "Substantial soil COS emission…")

At any rate, I'm providing a perspective here to help clarify your own structure, without expecting to include all or even any of this in your introduction.
**Response:** Thank you for the nice perspective. One of the benefits of the open discussion format is that this is now available to the public. Our paper is not a review, and luckily we have the 2018 Whelan review out, but we now checked and adjusted our citations are consistent with this narrative, including the addition of the recent reference of Kaisermann et al 2018.
* * *
There are many studies that have furthered our understanding of OCS soil exchange. For your agricultural study site, you could focus the discussion on the curious problem of high OCS production in agriculturally managed soils. The progression of our understanding of soil OCS exchange moves forward like this:

(1) At first, wetlands are considered a huge source of OCS, though most of the data collected from pre-1992 have methodological problems detailed in (Castro and Galloway, 1991). OCS production is linked to redox potential. Our set of valid wetlands measurements to date are summarized in Whelan et al. (2018) Figure 4.

(2) Kesselmeier et al. (1999) performs the first thorough set of measurements on a single, oxic, agricultural soil and finds only OCS uptake with a soil moisture and temperature optimum. Adding a carbonic anhydrase inhibitor reduced uptake. This study produced a model of soil OCS exchange with a maximum of 10 pmol/m2/sec and suggested that oxic soils are entirely a sink, probably biotic. This idea was revisited with 4 soils in Van Diest and Kesselmeier (2008).

(3) Watts (2000) is a review paper that clarified the thinking of the time: anoxic soils are a large OCS source and oxic soils are a small sink.

(4) This view was challenged in the literature when Maseyk et al. (2014) dragged a QCL laser into a wheat field in Oklahoma. They installed an automated soil chamber which enabled near continuous observations of oxic soil OCS fluxes in the field. I think the Liu paper was a lab-based experiment, and not enough of a big deal was made over how surprising it was for dried rice paddy soil to produce OCS.

(5) The authors of the Maseyk study sent me a soil sample from the Oklahoma site, which I did a series of laboratory incubations with. I used a GC/MS, the more traditional measurement approach, to confirm that the Maseyk findings weren't the result of some sort of methodological problem. We sterilized the soil and subjected it to full spectrum light, and demonstrated that OCS production was linked to both light and temperature and was probably abiotic (Whelan and Rhew, 2015).

(6) We then got soil samples from many, many sites and repeated the approach with a QCL in Whelan et al. (2016). We showed that air-dried soils in all biomes except deserts exhibit net OCS production, increasing exponentially with temperature. If one subtracts this curve from measurements made at other soil moistures, a curve resembling the initial Kesselmeier (1999) model can be recovered. This showed that abiotic OCS production and biotic consumption were occurring, probably in most soils. No one knew at this point why agricultural soils had such large magnitude emissions with temperature.

(7) Kitz et al., (2017) was the first field study to make conclusions about the effect of light in the field. It should be noted that this was an agriculturally managed soil, not a natural grassland. This is something to keep in mind in your discussion of it (L293/4).

(8) Finally, Kaisermann (2018) showed that nitrogen suppresses OCS consumption, providing a path to solve the mystery of why some agricultural soils appear to have such large OCS production compared to other oxic soils.
* * *
The discussion on L283 to L326 could be improved, too. In particular, the role of carbonic anhydrase is introduced multiple times. Also, I invite you to revisit the solubility/hydrolysis figures in Whelan et al., (2018) and calculate the proportion of OCS that might be dissolved then lost to hydrolysis considering your soil moisture content and OCS within-soil concentrations. The mention of the CO and MgSO4 is an interesting idea that I wish we could explore more. While I know that your QCL is capable measuring CO, the ambient standards are tricky to maintain. Was CO measured? Is it possible to pull out any interesting relationships with CO concentration and temperature? Related, were the water vapor measurements mentioned in L79 calibrated? If so, how? It doesn't look like you used these, except indirectly for a water vapor correction on the other two gases.

**Response:** The relevant part and the CA issue was streamlined. We appreciate some suggestions for the follow-up studies (unfortunately, we were not geared for CO

measurements, and water vapor fluxes and soil moisture are tricky business and part of separate study).

Section 4.2 focuses on SRU values. Often the point of understanding soil OCS exchange is to "correct" ecosystem level exchange so that a more accurate GPP estimate can be made. A possible motivation for comparing soil OCS exchange and soil $CO_2$ fluxes would be to try and estimate OCS soil fluxes from soil respiration model output. Otherwise, I'm not clear on what we gain by using SRU. In short, more justification is needed for this section.

**Response:** Indeed, this is part of the motivation. This overlaps with comment of Ref 1 was addressed also there by better introducing the point up front in the Introduction.

Overall, this is an important and interesting study. Your ability to predict most OCS soil fluxes using only two variables is a great achievement and also heartening for other modeling efforts. Soil profiles would be very useful to support process-based soil OCS exchange modeling. I look forward to seeing this paper in its final form.

**Response:** Thanks.

[revised manuscript text omitted]

---

## Author Response (AR2)

12 March 2019

Dear Editor,

Thank you for your positive Decision on our paper. As requested, we now made all the additional technical revisions you indicated and hope you will find the paper ready for publication in ACP.

Sincerely yours,

Fulin Yang and Dan Yakir on behalf of all co-authors.
* * *
**Editor comments:**

Lines 17-20: "The spatial heterogeneity in soil COS exchange indicated net uptake below and between trees of up to -4.6 pmol m-2 s-1, and net emission in sun exposed soil between rows, of up to +2.6 pmol m−2 s−1, with a mean uptake value of -1.10 ± 0.10 pmol m−2 s−1." Negative uptake, as stated in the text, is emission. Thus, this sentence needs revision. My suggestion is "The spatial heterogeneity in soil COS exchange indicated net uptake below and between trees of up to 4.6 pmol m-2 s-1, and net emission in sun exposed soil between rows, of up to 2.6 pmol m−2 s−1, with overall mean uptake value of 1.1 ± 0.1 pmol m−2 s−1."

**Response:** We agree with and revised as suggested.

Line 27: "-0.37". The second decimal in this value is not significant (does not contain information) as the uncertainty of the value is 0.3. Thus, I recommend expressing it here as -0.4 ± 0.3 (see also my comments below).

**Response:** Revised as suggested.

I have several changes to the representation of significant digits of numeric values, These are listed below.

**Response:** We accepted all these changes to the significance digit, and updated in the text and also in the abstract as suggested.

Line 20: $1.10 \pm 0.10$ pmol m−2 s−1 should be written as $1.1 \pm 0.1$ pmol m−2 s−1 as second decimal is as uncertain to be insignigicant.

Line 23: $-1.02 \pm 0.26$ pmol m-2 s-1 should be expressed as $-1.0 \pm 0.3$ pmol m-2 s-1.

Line 180: "$-1.10 \pm 0.10$ pmol m-2 s-1". As mentioned above, should be written as $1.1 \pm 0.1$ pmol m−2 s−1.

Line 183: "$-3.00 \pm 0.10$". This should be written as $-3.0 \pm 0.1$.

Line 252: "$4.40 \pm 0.29$ mm2 s-1". This should be written as $4.4 \pm 0.3$ mm2 s-1

Line 254: "$-1.02 \pm 0.26$ pmol m-2 s-1" should be $-1.0 \pm 0.3$ pmol m-2 s-1.

Line 255: "$-1.10 \pm 0.10$ pmol m-2 s-1" should be $1.1 \pm 0.1$ pmol m−2 s−1.

Line 270: "$-0.37 \pm 0.31$" should be written as $-0.4 \pm 0.3$. **This should be reflected in the value reported in the abstract**.

Line 282: "$(+0.53 \pm 0.66)$" should be written as $(+0.5 \pm 0.7)$.